# Unusual Application of Insect-Related Evidence in Two European Unsolved Murders

**DOI:** 10.3390/insects12050444

**Published:** 2021-05-13

**Authors:** Francesco Introna, Cristina Cattaneo, Debora Mazzarelli, Francesco De Micco, Carlo Pietro Campobasso

**Affiliations:** 1Institute of Legal Medicine, Department of Interdisciplinar Medicine, University of Bari “Aldo Moro”, Policlinico Hospital, 70124 Bari, Italy; francesco.introna@uniba.it; 2Institute of Legal Medicine, Department of Health and Biomedical Sciences, University of Milan, 20133 Milan, Italy; cristina.cattaneo@unimi.it (C.C.); debora.mazzarelli@guest.unimi.it (D.M.); 3Institute of Legal Medicine, Department of Experimental Medicine, University of Campania “Luigi Vanvitelli”, 80138 Naples, Italy; francesco.demicco@gmail.com

**Keywords:** forensic entomology, cold cases, insect evidence, hair evidence

## Abstract

**Simple Summary:**

Proper collection and analysis of physical evidence including insects, bloodstains, or any other material can be of probative value in a court of law. This is the first casework where hairs were involved as insect-related evidence. Hairs constitute important categories of trace evidence as they can provide useful information for an association between a suspect and a crime scene or a suspect and a victim. Two “cold cases” occurred in two different European countries in which the trace evidence relating to insects was the last piece of a complex puzzle useful for the conviction of the perpetrator.

**Abstract:**

Insect-related evidence must be considered of probative value just as bloodstains, fingerprints, fibers, or any other materials. Such evidence if properly collected and analyzed can also provide useful details in the reopening of old unsolved murders, also called “cold cases”. This paper presents the case of two murders that occurred in two different European countries and remained unsolved for years. The remains of a girl found in Italy 17 years after her disappearance helped to solve a murder that occurred in Britain 8 years prior. The cases were unexpectedly linked together because of the similarities in the ritualistic placing of strands of hair and connections with the suspect. The trace evidence relating to insects and hairs played a relevant role in the conviction of the perpetrator. In Italy, the defense raised the doubt that the strands of hair found nearby the skeletal remains could be the result of insect feeding activity and not the result of a cut by sharp objects. Therefore, it was fundamental to distinguish between sharp force lesions and insect feeding activity on hair. This unusual application of insect-related evidence clearly emphasizes the importance of an appropriate professional collection and analysis of any physical evidence that could be of robust probative value.

## 1. Introduction

The major goal of medicocriminal entomology is the determination of time, cause, and manner of an investigated death [1]. The most frequently requested task is the estimation of the time since death, better known as the minimum postmortem interval (PMI min) [2]. However, new insights into and achievements of carrion entomology and ecology have increased the opportunities for the use of insects in death investigations and their accuracy [3,4,5].

Several useful elements can be inferred from the study of insects found on the cadaver or nearby. The correct analysis and interpretation of insect evidence can be crucial in any death investigation, especially when dealing with badly decomposed or skeletonized human remains. Insect-related evidence can provide valuable information to answer questions concerning the use of drugs, child and elderly neglect, sexual abuse, cadaver transfer and concealment of the remains following death, victim identification, determination of specific sites of antemortem trauma, and postmortem artefacts on a body and at the death scene [6,7]. The final aim of any criminal investigation is to provide a strong basis for an association between a suspect and a crime scene or a suspect and a victim.

Therefore, insect-related evidence must be considered of probative value just as bloodstains, fingerprints, hairs, fibers, or any other biological materials. In the recent past, several papers have highlighted the crucial role of correct procedures in sampling and storing entomological evidence [8,9,10]. If collected properly, such evidence can also provide useful details in the reopening of old unsolved murders and in the presentation of important evidence to the court [11].

This paper presents the case of two murders that occurred in two different European countries and remained unsolved for years. The murders were unexpectedly linked together because of the similarities in the ritualistic placing of strands of hair and connections with the suspect. Hair evidence and its postmortem infestation by insects played a relevant role in the conviction of the perpetrator. The remains of a girl found in Italy 17 years after her disappearance helped to solve a murder that occurred in Britain 8 years prior.

## 2. Case History

On 17 March 2010, the mummified remains of a 16 year old female were discovered hidden under curved tiles in the darkest corner of a church’s loft in Southern Italy. The victim was identified as that of a teenage girl who disappeared from her home 17 years before, on 12 September 1993. The cause of death was massive blood loss due to multiple stab wounds. At autopsy, the signs of multiple stab wounds (15 in total) were found on the skeletal remains. Large amounts of *Diptera puparia* (mostly parasitized) and cast pupal skin of clothes moths (Lepidoptera) were present on the body and nearby the corner of the church’s loft where the victim was dragged and concealed. Among the *Diptera* species, Sarcophagids, Phoridae, Fanniidae, Muscids of *Muscina stabulans*, and Calliphorids of *Calliphora vicina*, *Lucilia sericata*, and *Chrysomya albiceps*, on which the clothes moths fed, were found (Figure 1). Among the clothes moths, *Tinea pellionella* and *Tinea bisseliella* were found. The insect species collected were those that were active during the fall, when the victim disappeared. Physical evidence collected at the death scene was some light-brown strands of hair, perfectly and squarely cut, near the skeletonized hands (Figure 2).

The murder case showed some coincidences with a similar case that occurred in England 8 years prior. A 48 year old seamstress was found dead in her apartment on 12 November 2002. The woman was beaten with a hammer-type object before her throat was cut and body mutilated. She was also stabbed several times and strands of hair were placed in both of her hands. The lock of hair in the victim’s left hand was cut from her own head, but that in her right hand was not hers. The main suspect in this murder was the victim’s neighbor. He was a 39 year old Italian male who arrived in England several years before from a small town in southern Italy, where the teenage girl went missing. Although the neighbor initially became the main suspect, the British police was unable to collect sufficient evidence for prosecution. After the discovery of the Italian girl in 2010, the cases were linked together because of the similarities in the ritualistic placing of strands of hair and connections with the suspect. The manner in which the English woman was murdered was considered a trademark of the suspect, and the killing was linked to the murder of the adolescent girl in Italy. Two months after the discovery of the Italian girl in 2010, the suspect was arrested in England and charged with the murder of the British woman. In June 2011, he was found guilty by the English jury and sentenced to spend the rest of his life in prison. On October 2014, he was also condemned by the Italian Supreme Court to 30 years in prison for the murder of the 17 year old teenage girl.

## 3. Insect-Related Evidence

At the trial, the perpetrator consistently denied any involvement in both cases. However, under cross-examination he admitted to having a hair fetish and cutting the hair of girls occasionally met on the street. He liked to touch and smell the hair of girls without planning cutting episodes. According to such admission, the two murders were referred as hair fetish murders. Hair fetishism or trichophilia is a paraphilia, which is a sexual perversion, a condition characterized by abnormal sexual desires. In the case of hair fetishism, seeing or touching hair is particularly erotic and sexually arousing for a person [12]. During the trial, several women reported having their hair cut by an unknown man. One of them described finding something white and sticky in her hair afterwards. Another man described seeing the suspect sitting behind a woman on a bus with her hair in his hands.

In Italy, the defense raised the doubt that the strands of hair found nearby the skeletal remains were not cut by sharp objects but could be the result of insect feeding activity on hair. Therefore, it was fundamental to distinguish between sharp force lesions and insect feeding activity on hair. In order to study if the features in hair lesions were caused by traumatic/sharp forces or by insects, hair samples were subjected to blunt and sharp force trauma and used as pabulum for common clothes moth (*Tineola bisselliella* Lepidoptera, Tineidae) and carpet beetles (*Anthrenus* sp., Colepoptera, Dermestidae) [13]. These species colonize bodies in the late stages of decay when the food source is completely dry [14,15], feeding not only on natural fiber such as hairs, but also on clothing [16]. Artefacts on hairs caused by the feeding activity of moths and beetles were studied by two of the coauthors in a published manuscript [13].

The hair samples examined by stereomicroscopy and scanning electron microscopy (SEM) showed clear differences in hair lesions depending on the type of trauma. They can be summarized as follows: (1) irregular edges without striations and concavities, but with hair elements on different levels in hair locks manually broken; (2) regular edges with parallel striations (similar to marks from sharp weapons observed on bone and cartilage) and hair elements on the same plane in hairs cut by a knife, with the same features but of oval shape due to the compression of the two edges of scissors in hairs cut by scissors; (3) irregular edges with concave shape lesions and cocoons in hair locks used as pabulum for insects [13].

The strands of hair found close to the hands of the victim showed regular edges of the hair lesions with parallel striations of oval shape consistent with hairs cut by scissors. These findings and the results of the experimental study were considered of probative value. The hair lesions distributed on the same plane were clearly cut and not the result of gnawing activity easily distinguishable from breaking and tearing. In the Italian case, the hypothesis of insect feeding on hair was ruled out.

## 4. Discussion

Hair evidence is one of the most common pieces of physical evidence applied in criminal investigations [17]. The forensic analysis of hair evidence can be extremely valuable in demonstrating that there may have been an association between a suspect and a crime scene, a weapon, or a victim [18,19]. Comparing strands of hair under a microscope can provide robust corroborative evidence to establish these associations as hairs constitute important categories of trace evidence [19]. Microscopical analysis of hairs can give significant information regarding the ancestry, the body area the hair came from, and whether or not the hair was artificially treated or damaged. Toxicological and genetic analysis of hair can also provide useful information from the identification of the hair donor to the detection of drugs and pharmaceuticals and their chronology of intake, especially in drug-facilitated crimes and sexual assaults [20,21]. To the best of our knowledge, this is the first forensic case where hair evidence was related to insect artefacts due to feeding activity on hair.

At a death scene, fly and larval activity can produce several different types of artefacts: from fly specks due to regurgitation and defecation to floor and wall stripes due to post-feeding larval dispersal [22]. Other modifications of the bloodstain pattern are represented by transfer stains due to the migration of insects from a blood pooling [23]. However, *Diptera* larvae can also produce relevant modifications of superficial injuries on the human body. Stab wounds and gunshot injuries can be easily enlarged, distorted, or made unrecognizable by the feeding larval activity at the injury site [24]. Larvae actively feeding on soft tissues can destroy the residual part of the physical evidence still present on the body. Therefore, corpses heavily colonized by larval masses should be considered a matter for urgent forensic pathology examination, and the autopsy should be performed as soon as possible [6,24].

Artefacts made by insects that arrive late on the body are also reported. Dominant taxa in the late stages of decomposition are commonly species of Coleoptera and Lepidoptera such as the clothes moths (Tineidae), very common in mummies and skeletonized bodies [14,15]. These species colonize bodies when the remnants of soft tissues are really dry, feeding on them, as well as on hairs and other natural fibers such as those of clothing [16]. Coleoptera and Lepidoptera can also feed on *Diptera* puparia.

Puparia consist largely of chitin, a long, complex polysaccharide polymer composed of *n*-acetylglucosamine that is similar to keratin in human hair. Keratin and keratin-associated proteins are the major structural components of human hair fiber. Human hairs are proteinaceous shafts with circular or elliptical cross-sections with three morphologically distinct regions in each hair (medulla, cortex, and cuticle) [19]. They can be considered as fiber-reinforced composites consisting of crystalline intermediate filaments embedded in an amorphous protein matrix [25]. The strength and robustness of keratin is derived from tightly packed filaments containing a high degree of disulfide bonding, which confers rigidity and chemical resistance.

The structure and mechanical properties of puparia and human hair explain why sclerotized puparia and hairs do not degrade much, which allow samples to survive for thousands of years [19,26]. Fly puparia and human hairs have been found in ancient graves from Egypt to South America, long after soft tissues have disappeared [27,28]. *Diptera* puparia may represent useful evidence for reconstructing postmortem events in both forensic and archaeo-funerary contexts [29]. However, they may contaminate the forensic entomological evidence if they originate not from human remains but from animal cadavers or other decomposing organic material [30].

The taphonomic degradation process for teeth and bones is well studied, but the same cannot be said for human hair as very little is known about how environmental conditions may alter hair morphology [19]. A recent review [31] addressed the main factors that correlate with decompositional changes of hair. Temperature, environment, and microbes are the major factors affecting the decay of hair. In particular, humidity/aqueous conditions and warm temperatures appear to have a significant effect on the rate of hair decomposition [32,33].

Fungi and bacteria can break down hair; however, insects such as clothes moths and carpet beetles can also easily destroy such evidence as they feed on dried food sources such as natural fibers [34,35,36]. Many moths and beetles use keratin as a nutrient source. Moth larvae are common sources of damage of woolen textiles, and many Dermestidae larvae also consume hair. The gnawing insect activity usually leaves microscopical signs of erosions and concave lesions on the surface of the hair, but these signs were not observed in the hair samples found close to the skeletonized hands of the victim. In the Italian case, the absence of such microscopical features related to feeding insect activity were considered enough evidence to exclude hair damages due to postmortem infestation by insects as suggested by the defense. Therefore, the hypothesis of insect feeding on hair was ruled out. The hair samples showed only the marks produced from sharp force weapons such as those produced by the compression of the two edges of scissors. Such evidence was just a piece of the big puzzle presented by this international death investigation of unsolved and heinous crimes that occurred in England and Italy, but no less important and crucial for the indictment and final conviction of the suspect.

## 5. Conclusions

The mutually beneficial relationship between research and casework was largely discussed by Hall [37]. The unusual application of insect-related evidence to hair evidence clearly emphasizes the importance of an appropriate professional collection and analysis of any physical evidence that could be of robust probative value [38]. Differentiation of entomological activity, taphonomy, and sharp force trauma on hair can be crucial in the recovery and analysis of hair evidence.

The take-home message is to call the attention of forensic entomologists to hair examination in skeletonized or mummified bodies or in an advanced stage of decay. Forensic experts should cover every corner of the crime scene and treat each and every piece of evidence as vital. Investigative research has contributed to advancing knowledge of the basic biological and ecological processes of human and animal decomposition, increasing the chances for the admissibility of entomological evidence [39,40]. Because criminal investigations are context-dependent, any trace evidence can provide useful information about the event under investigation, and proper collection and preservation of physical evidence is mandatory [41]. The final aim of any forensic examination must be to provide statements based on objective scientific observation that can be of value in a court of law or for any interested party involved in criminal investigations [18].

## Figures and Tables

**Figure 1 insects-12-00444-f001:**
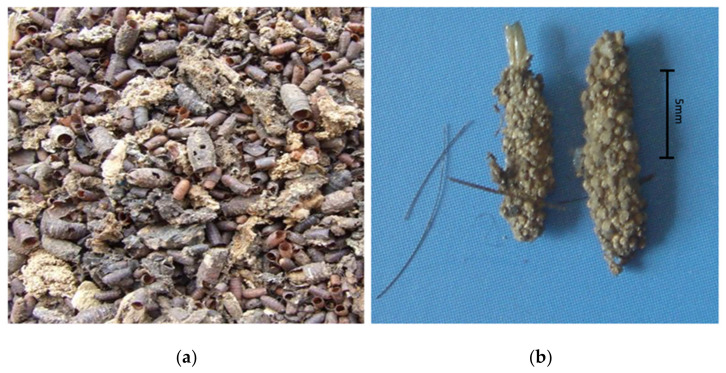
(**a**) Diptera puparia mostly parasitized; (**b**) clothes moth cases and hairs.

**Figure 2 insects-12-00444-f002:**
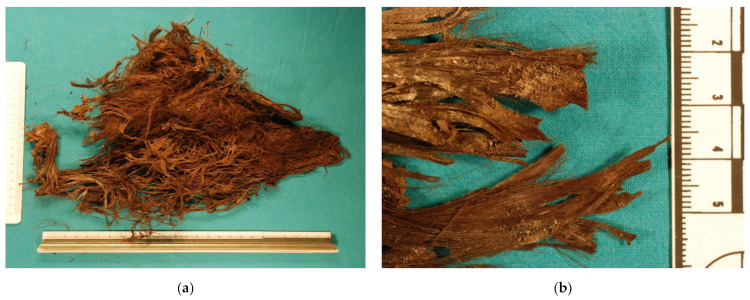
(**a**) Strands of hair found nearby the skeletal remains of a 16 year old female found in Italy 17 years after death; (**b**) close up of the lock of hair clearly cut by sharp force trauma (scissors).

## Data Availability

The data presented in this study are available on request from the corresponding author.

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
