# Peer review of "Unusual Application of Insect-Related Evidence in Two European Unsolved Murders"

_insects, 2021, doi:10.3390/insects12050444_

Round 1
Reviewer 1 Report
This was a very interesting case study. I have issues only with some of the sentence structure and grammar. I am sure these issues can be easily corrected.

Author Response
The Authors would like to thank Reviewer#1 for his helpful comments.
We appreciate the opportunity to improve the manuscript. Your suggestions and advices dealing with the sentence structure and grammar have been taken into consideration. English in the manuscript has been thoroughly checked and edited for language and form by a native English speaker.
Please, for details, have a look at the new version of the manuscript (in attachment) where all changes are highlighted in red color, so that changes are easily visible to the editors and reviewers. Thank You very much for the improvement made to the manuscript.
Reviewer 2 Report
This case report enlarges the area of application of forensic entomology.
Hair infestation by insects and its morphological findings have been overlooked.
This paper's case presentation is very simple, but straightforward.
Visual examinations for hair may play a role to explain whether the mechanisms of hair damages are mechanical or due to postmortem-infestation by insects.
This paper calls attention to hair examination even in the field of forensic entomology.
However, although I am not a native English speaker, I found several awkward English sentences.
I recommend authors utilize a professional English editing service to refine their manuscript.
I also found a possible error of taxonomic description.
Please refer to the following.
Line 61: "remains". Was the 16-year-old girl's body dismembered or fragmented? If not, "remain" would be better.
Line 61: "16-years-old" should be "16-year-old".
Line 62 and 63: "was identified for" is weird.
Line 69 and 70: "Among the Diptera species, Sarcophagids and Calliphorids like puparia of Calliphora vicina, Lucilia sericata, Chrysomya albiceps, and Muscina stabulans, on which clothes moths fed, were found."
-> L.sericata and C.albiceps are Calliphorids. M. stabulans is a Muscid, not a Sarcophagid.
Author Response
The Authors would like to thank Reviewer#2 for his comments.
1) We have really appreciated that Reviewer#2 found our case report interesting as it enlarges the area of application of forensic entomology. That is exactly the goal of the manuscript. This was also the reason why we have chosen a simple way for the case presentation. The purpose of the case report was to call attention of forensic entomologists to hair examination especially in skeletonized or mummified bodies and how it can be useful in a death investigation.
2) According to your comments, English in the manuscript has been thoroughly checked and edited for language and form by a native English speaker. In the revised version, several changes have been made in order to avoid awkward English sentences and misunderstanding on possible error of taxonomic description. Please, for details, have a look at the new version of the manuscript (in attachment) where all revisions are highlighted in red color, so that changes are easily visible to the editors and reviewers.
We really hope that the proposed revisions and comments be clear and acceptable. Thank You very much for your suggestions and the improvement made to the manuscript.
Reviewer 3 Report
This paper includes an application in forensic practice of knowledge of insect biology and behaviour but, unfortunately, does not provide new information important enough.
Forensic techniques used in this work is already explained and published by one of the authors (reference 12 in the text) and this paper not include new information or an exhaustive review in relation to insects in forensic practice.
This paper should be modified by providing more information, pictures… on the insects and evidences involved in the cases explained, which support and highlight the importance of the information presented. Otherwise the paper must be changed in a short communication.
Author Response
The Authors would like to thank Reviewer#3 for his comments.
1) We respect the concerns raised but we would like to inform Reviewer#3 of a notable disagreement between Reviewer#3 and the reports provided by other two Reviewers. Two Reviewers have found the case report interesting as it enlarges the area of application of forensic entomology. The purpose of the manuscript is to call attention of forensic entomologists to hair examination especially in skeletonized or mummified bodies and how it can be useful in a death investigation. This is a new information that two other Reviewers believe important enough.
2) It is true that two of the co-Authors already studied the mechanisms of hair damages and published the results of their research. We are aware of that as we respectfully cited this reference. However, in the article already published on 2015, no details were provided regarding the murders that were linked together because of the hair evidence. No details were reported about hair fetishism, how the trace evidence on hair was associated to artefacts by insects and how it was applied in the death investigation. The paper in revision also provides some information about the potential application of hair evidence in crime investigation and the similar mechanical properties of human hair and Diptera puparia which allow samples to survive for thousands of years.
3) Of course, we have taken into consideration your suggestion to provide more information and pictures. Some additional information about the insect community associated to the skeletal remains and a new image depicting Diptera puparia and clothes moth cases have been inserted. Unfortunately, we cannot provide images of the hair damages already published on Forensic Sci Med Pathol because of the copyright of Springer publishing company. However, the reader can easily find the morphological features of hair lesions on the cited article.
4) In the revised version, several changes have also been made in order to avoid awkward English sentences and misunderstanding on possible error of taxonomic description. English in the manuscript has been thoroughly checked and edited for language and form by a native English speaker. Please, for details, have a look at the new version of the manuscript (in attachment) where revisions are highlighted in red color, so that changes are easily visible to the editors and reviewers.
We really hope that the response to the concerns and the proposed revisions be acceptable.
Round 2
Reviewer 3 Report
N/A